# Waist–Calf Circumference Ratio Is Associated with Body Composition, Physical Performance, and Muscle Strength in Older Women

**DOI:** 10.3390/geriatrics10040103

**Published:** 2025-08-01

**Authors:** Cecilia Arteaga-Pazmiño, Alma L. Guzmán-Gurrola, Diana Fonseca-Pérez, Javier Galvez-Celi, Danielle Francesca Aycart, Ludwig Álvarez-Córdova, Evelyn Frias-Toral

**Affiliations:** 1Carrera de Nutrición y Dietética, Facultad de Ciencias Médicas, Universidad de Guayaquil, Guayaquil 090613, Ecuador; ceciliaarteagap@gmail.com (C.A.-P.); johnny.galvezc@ug.edu.ec (J.G.-C.); 2High Specialty Geriatric Care Unit, Hospital Civil de Guadalajara Fray Antonio Alcalde, Guadalajara 44200, Jalisco, Mexico; aguzman@hcg.gob.mx; 3Centro Universitario de Tlajomulco, Universidad de Guadalajara, Tlajomulco 45641, Jalisco, Mexico; 4Carrera de Nutrición y Dietética, Facultad de Ciencias de la Salud, Universidad Católica de Santiago de Guayaquil, Guayaquil 09014671, Ecuador; diana.fonseca@cu.ucsg.edu.ec; 5Facultad de Ciencias de la Vida, Escuela Superior Politécnica del Litoral (ESPOL), Campus Gustavo Galindo Km. 30.5 Vía Perimetral, P.O. Box 09-01-5863, Guayaquil 090902, Ecuador; aycartdanielle@gmail.com; 6Facultad de Ciencias de la Salud, Universidad de Las Américas (UDLA), Quito 170513, Ecuador; 7Escuela de Medicina, Universidad Espíritu Santo, Samborondón 0901952, Ecuador; evelynft@gmail.com; 8Division of Research, Texas State University, 601 University Dr, San Marcos, TX 78666, USA

**Keywords:** body composition, muscle mass, calf circumference, waist circumference, older women

## Abstract

**Background:** The waist–calf circumference ratio (WCR) is an index that combines waist and calf circumference measurements, offering a potentially effective method for evaluating the imbalance between abdominal fat and leg muscle mass in older adults. **Objective:** To assess the association between WCR and indicators of body composition, muscle strength, and physical performance in community-dwelling older women. **Methods:** This was a cross-sectional study involving 133 older women (≥65 years) from an urban-marginal community in Guayaquil, Ecuador. The WCR was categorized into quartiles (Q1: 2.07–2.57; Q2: 2.58–2.75; Q3: 2.76–3.05; Q4: 3.06–4.76). Body indicators included fat-free mass (FFM), skeletal muscle mass (SMM), appendicular muscle mass (ASM), appendicular muscle mass index (ASMI), visceral fat (VF), fat mass (FM), and fat mass index (FMI). Handgrip strength (HGS) and the Short Physical Performance Battery test (SPPB) score were used to assess muscle strength and function, respectively. **Results:** The median age of the participants was 75 [IQR: 65–82] years. The mean WCR was 2.92 ± 0.93. Statistically significant associations were found between WCR and VF (*p* < 0.001), WCR and SMM (*p* = 0.039), and WCR and ASM (*p* = 0.016). Regarding muscle function, WCR was associated with HGS (*p* = 0.025) and SPPB score (*p* = 0.029). **Conclusions:** A significant association was observed between WCR and body composition, and muscle strength and function in older women.

## 1. Introduction

Life expectancy has increased globally, with the World Health Organization (WHO) projecting that by 2030, one in six people will be aged 60 or older [1]. Furthermore, between 1950 and 2020, global life expectancy increased from 46 to 73 years [2], and by 2050, the global population of older adults is expected to double, reaching approximately 1.6 billion [1]. The fastest growth of the older population is occurring in Africa, Latin America, the Caribbean, and Asia, and it is estimated that by 2050, 80% of people aged 65 and older will reside in these regions [3]. Women, in particular, tend to live longer than men, with the largest gender gap in life expectancy—approximately 6.5 years—observed in Latin America and the Caribbean [4].

Aging is a natural and progressive process that begins in early adulthood and is marked by a gradual decline in physiological functions [5,6]. It is commonly accompanied by changes in body composition, including increased fat accumulation and loss of muscle mass [7,8,9]. Biological differences in the aging processes of men and women are driven mainly by hormonal changes. These hormonal changes, particularly during menopause, increase the risk of conditions like osteoporosis or Alzheimer’s in women [10,11,12]. In addition, these changes can impact the overall health and functional capacity of older adults [13,14], especially in women who exhibit lower functional capacity compared to men [15].

Accurate assessments of body composition for detecting adverse changes that may elevate health risks in older adults are essential [16]. However, advanced body composition analysis methods, such as bioelectrical impedance analysis, are not feasible in certain low-resource or community-based settings due to the equipment cost, maintenance needs, and lack of trained personnel [17].

Given these limitations, simple anthropometric measures such as waist circumference (WC), an indicator of central adiposity, and calf circumference (CC), a proxy for peripheral muscle mass, are gaining interest as alternative tools for evaluating body composition in older populations, especially in resource-limited settings [18]. The waist-to-calf circumference ratio (WCR), which integrates these two measures, has emerged as a potentially valuable index for identifying sarcopenic obesity and other age-related health conditions [19].

Recent studies have linked a higher WCR and WC, along with a lower CC, to increased risks of cognitive decline, frailty, and mortality in older adults [20,21]. These findings highlight the potential clinical utility of the WCR in identifying individuals at higher risk of adverse health outcomes [22]. However, the evidence remains limited, particularly regarding its application in older women in Latin America—a population that is expected to grow substantially in the coming decades.

Therefore, this study aims to examine the association between WCR and indicators of body composition, muscle strength, and physical performance in older women in Guayaquil, Ecuador. This relatively novel metric aims to determine its potential utility in assessing age-related changes and health risks in this population.

## 2. Materials and Methods

### 2.1. Participants

An observational cross-sectional study was carried out in a community-dwelling center with older women (≥65 years) in urban-marginal areas of Guayaquil, Ecuador, from November 2019 to December 2020. The sample was selected through non-probabilistic sampling by applying inclusion and exclusion criteria. The inclusion criteria were the following: women who agreed to participate voluntarily in the study by signing an informed consent form. The exclusion criteria were the following: institutionalized individuals, known severe cognitive impairment, functional dependence, current cancer, chronic obstructive pulmonary disease, musculoskeletal diseases, and amputations of the lower or upper limbs. Figure 1 shows a flowchart of the recruitment process.

The participants completed a self-reported standardized questionnaire to evaluate their socioeconomic and clinical characteristics. The socioeconomic variables included age, ethnicity (Mestizo, Afro-Ecuadorian, Caucasian, or Indigenous), marital status (single, married, widowed, or divorced), and education level (none, elementary, high school, or university). Comorbidities were assessed on the basis of chronic medical conditions such as type 2 diabetes, hypertension, dyslipidemia, and arthritis. These comorbidities were excluded to minimize potential confounding effects on muscle function and mobility assessments, particularly in variables such as gait speed, chair stands, and handgrip strength.

### 2.2. Anthropometric Measures

Trained researchers performed the anthropometric measurements. A standard stadiometer and an electronic weighing scale were used to measure the participants’ height and weight. The results were used to calculate their body mass index (BMI) using the standard formula weight (kg)/height (m)^2^, and the standard interpretation BMI ranges were used for the study.

#### 2.2.1. Waist Circumference Assessment

Waist circumference was assessed using a non-stretchable pliable metal tape. The participants were asked to stand with their feet aligned closely together. WC was measured at the narrowest area between the ribs and hips at the end of a normal expiration. Abdominal obesity was defined as WC ≥ 88 cm [23].

#### 2.2.2. Calf Circumference

The measurement of CC was conducted while the participants were in a sitting position, with their feet resting on the floor and their knees and ankles at a right angle. These measurements were obtained at the point of greatest circumference, which was not covered by clothes, on the right calf, perpendicular to its length, without applying pressure to the underlying skin layers. The measurements were rounded to the nearest 0.1 cm.

Then, WCR was calculated by dividing WC by CC. The participants were categorized into groups according to WCR quartile scores as follows: Q1: 2.07–2.57; Q2: 2.58–2.75; Q3: 2.76–3.05; and Q4: 3.06–4.76. The quartiles of WCR are used as representative groups because there are no validated clinical cutoff points for WCR in older women in Latin America; previous studies used distribution-based cutoffs when normative values were lacking [24,25].

### 2.3. Body Composition Analysis

Body composition compartments were measured using a Multifrequency Segmental Body Composition Analyzer (InBody 270 DSM-BIA^®^): fat-free mass (FFM), skeletal muscle mass (SMM), appendicular muscle mass (AMM), appendicular muscle mass index (ASMI), visceral fat (VF), fat mass, (FM), and fat mass index (FMI). AMM was calculated using the Sergi formula [26]. The participants were advised not to eat or drink 4 h before the test, consume any caffeine beverage or alcohol within 12 h of the test, use diuretic medication, or perform exercise 12 h before the test, and they were encouraged to evacuate urine before the assessment was performed.

### 2.4. Muscle Function Assessment

Muscle function was assessed using the Short Physical Performance Battery (SPPB), which includes three physical performance measures: standing balance, repeated chair stands, and gait speed [27]. The balance evaluation consisted of hierarchical tasks involving side-by-side, semi-tandem, and full-tandem stands. During the repeated chair stand test, the participants were timed while performing five sit-to-stand repetitions. Gait speed was assessed by timing the participants as they walked 2.44 m at their usual pace.

Each assessment was graded on a scale from 0 (indicating an inability to complete the task) to a maximum of 4 points (representing the highest level of performance on the test). The overall SPPB score ranges from 0 (indicating the poorest performance). Performance in these tests is based on the following four categories: 0–3 points (indicating disability/very poor performance), 4–6 points (indicating poor performance), 7–9 points (indicating moderate performance), and 10–12 points (indicating good performance).

### 2.5. Muscle Strength

Muscle strength was assessed through handgrip strength (HGS) using a Hand Dynamometer (Jamar Plus model 5030 J1, Sammons Preston Rolyan, Bolingbrook, IL, USA) with an accuracy of over 99% [28]. HGS was assessed in both hands, regardless of which one was dominant. The participants were verbally instructed to grip the instrument and exert maximum handgrip strength. All tests were conducted while standing, with both arms hanging at the sides and the dynamometer facing the evaluator. The highest value recorded between both sides was used, and individuals rested for at least one minute between trials of the same hand. Dynapenia was identified when HGS was < 16 kg [29].

### 2.6. Other Variables

Nutritional status was assessed using the Mini Nutritional Assessment Short Form (MNA-SF) [30]. Frailty was measured using the FRAIL scale [31], and sarcopenia risk was assessed via the SARC-F questionnaire [32].

### 2.7. Statistical Analysis

Data analysis was performed using IBM SPSS Statistics (version 25.0; IBM, Chicago, IL, USA). A descriptive analysis of categorical variables was performed by examining their frequencies and distributions. Continuous variables are reported as mean and standard deviation or median and interquartile range [IQR]. Normal distributions were evaluated using the Kolmogorov–Smirnov test. For the bivariate analysis, the Kruskal–Wallis test was performed, and the Bonferroni correction was applied for the Kruskal–Wallis test.

Confounding variables were those that showed a significant association with the dependent variable and with WCR in bivariate analysis. In this study, age, BMI, nutritional status, frailty, and risk of sarcopenia were determined as confounding variables.

The relationship between WCR and indicators of body composition, function, and muscle mass was analyzed using Spearman’s correlation.

Linear and logistic regression models were constructed, adjusting for age, BMI, nutritional status, frailty, and risk of sarcopenia. For each model, the following were evaluated: coefficients (β), standard errors, 95% confidence intervals, z or t statistics, *p*-values, and (for logit models) odds ratios. Additionally, adjusted R^2^ (for linear regressions) and AUC (for logistic models) were reported. For all analyses, a *p*-value < 0.05 was considered statistically significant.

### 2.8. Ethical Considerations

This study was approved by the Ethics Committee for Research in Humans of the “Hospital Clínica Kennedy” in Guayaquil, Ecuador (CEISH No: HCK-CEISH-19-0038, 21 June 2019) and was conducted according to the guidelines of the Declaration of Helsinki [33]. All participants were informed about the study, its objectives, and the instruments used, and subsequently provided written consent. During the study, the data collected from the participants were de-identified.

## 3. Results

### 3.1. Characteristics of the Subjects

A total of 133 older women participated in this study. The median age of the participants was 75 [IQR: 65–82] years. Table 1 summarizes the characteristics of the studied population. Over half of the sample was mestizo (69.20%). Approximately 30.10% of the participants were single, and 54.90% reported elementary education.

Regarding nutritional status, 45.90% of the individuals showed nutritional risk. Furthermore, 18.85% of the participants were frail, and 47.40% presented with sarcopenia risk. Regarding waist circumference, 67.70% exhibited abdominal obesity.

### 3.2. Waist-to-Calf Ratio (WCR)

The median WC was 92.37 [IQR: 90.50–94.24] cm, the median CC was 33.20 [IQR: 29.85–35.90] cm, and the mean WCR was 2.92 ± 0.93.

### 3.3. Body Composition

The median FFM was 25.80 [IQR: 20.95–30.40] kg, the median SMM was 14.10 [IQR: 10.55–16.80] kg, the median ASM was 14.25 [IQR: 12.31–15.93] kg, and the median ASMI was 6.52 [IQR: 5.91–7.27] kg/m^2^. Furthermore, body fat was 25.80 [IQR: 20.95–30.40] kg, body fat index was 11.60 [IQR: 9.80–13.90] kg/m^2^, and visceral fat was 2.80 [IQR: 2.40–3.40] l.

### 3.4. Muscle Strength

The median HGS was 13 [IQR: 10.00–18.00] kg, and 60.90% of the participants had dynapenia.

### 3.5. Muscle Function

The median SPPB score was 7 [IQR: 4.00–8.00]. A total of 45.10% of the participants showed moderate physical performance, and 26.80% exhibited poor performance. Figure 2 shows a categorical analysis of dynapenia and muscle function according to the WCR quartiles.

### 3.6. Association Between WCR and Body Composition, and Muscle Strength and Function

Table 2 presents the differences of indicators of body composition, muscle function, and strength according to the WCR quartiles. Participants in the highest WCR quartiles exhibited significantly lower FFM, SMM, and ASM (*p* < 0.05) compared with participants in the lowest quartiles. However, only visceral fat was significantly higher as the quartiles increased (*p* < 0.001). Figure 3 shows the boxplots of SMM, visceral fat, and SPPB score according to WCR quartiles.

WCR is significantly and negatively associated with ASM (ρ = 0.198, *p* = 0.022) and HGS (ρ = –0.235, *p* = 0.006). In addition, there is a positive and significant association between WCR and visceral fat (ρ = 0.387, *p* < 0.001). The other variables show associations in the expected direction but do not reach significance (Table 3).

### 3.7. Regression Model Results

The results of the multiple linear regression for HGS show that a 1-unit increase in WCR was associated with a 3.625 kg decrease in HGS, independent of age, BMI, nutritional status, frailty, and sarcopenia risk (*p* = 0.026). The model explained 36.8% of the observed variability in HGS (adjusted R^2^ = 0.368).

Each 1-unit increase in WCR increases the odds of having dynapenia by ~140% (OR = 2.398; 95% CI: 1.271–4.523; *p* = 0.007). Figure 4A shows the ROC curve for dynapenia (HGS < 16). The extended logistic regression model for low SPPB showed that WCR is no longer significant in predicting SPPB < 7 (*p* = 0.389), as shown in Figure 4B. Age remains the only significant predictor (each additional year increases the odds of a low SPPB score by 6.4%). The optimal WCR threshold to discriminate SPPB < 7 is ≈2.87 (balanced sensitivity and specificity).

### 3.8. Post Hoc Analysis

Participants in Q4 had lower HGS compared to those in Q1 (mean difference = 3.82 kg), with a moderate to large effect size (Cohen’s d = 0.672; r = 0.318). Regarding SMM, a comparison between Q1 and Q4 showed a moderate effect size (Cohen’s d = 0.483; r = 0.235), although it did not reach statistical significance (*p* = 0.066).

Post hoc comparisons using Bonferroni-corrected Mann–Whitney U tests revealed significantly higher visceral fat in Q4 compared to Q1 (U = 207.0, *p* < 0.001), with a large effect size (r = 0.533). Likewise, the participants in Q4 had significantly lower SPPB scores than those in Q1 (U = 201.0, *p* < 0.001; r = 0.542) and Q2 (U = 289.0, *p* = 0.018; r = 0.417), indicating moderate to large effects.

## 4. Discussion

This study aimed to investigate the associations between WCR and body composition, muscle function, and strength in older women. The findings from this study showed that WCR was significantly related to indicators of muscle mass, function, and strength, as well as visceral fat. Moreover, the differences in the indicators evaluated were higher in the top quartiles of WCR.

The sample had a mean BMI in the overweight range, which underscores the clinical relevance of using WCR to detect unfavorable body composition patterns. Overweight older adults may have adiposity excess while also presenting with reduced muscle mass, a condition often undetected by BMI alone. Therefore, the observed associations between WCR and muscle strength and function in this population support the utility of WCR as a more sensitive marker of age-related health risks compared to BMI.

WCR has been proposed as an indicator for assessing obesity, muscle health, and cardiometabolic risks [25,34,35]. Recently, Güne et al. suggested that WCR may be useful for the diagnosis of sarcopenic obesity, as well as a valuable metric to provide insights into an individual’s body composition. In this sense, these authors proposed a cutoff point of 2.94 for the diagnosis of sarcopenic obesity in older adults [19], which is similar to the mean WCR of 2.92 in this population.

Regarding body composition, most indicators of muscle mass showed a correlation with WCR quartile, highlighting the usefulness of WCR quartiles as an indicator of reduced muscle mass [36,37]. On the other hand, among the indicators of fat, only visceral fat was associated with WCR. This result could be explained by the direct relationship between the distribution of fat in the abdomen, particularly visceral fat, and waist circumference [38,39]. Thus, total body fat and body fat index appear to be less accurate for identifying visceral fat, and therefore, they seem to be uncorrelated with WCR.

Simple, accessible proxy metrics for sarcopenia in older adults are needed in low economic settings where there is no access to advanced equipment [40]. In a sample of 1500 older adults, Mendes et al. reported that a lower CC, as well as a higher WC, was associated with low HGS [41]. These results are consistent with this study, in which participants showed lower handgrip strength in the upper quartiles of WCR. This relationship could be explained by the predictive value of CC for HGS [42]. These results highlight the close relationship between anthropometric indicators of muscle mass and HGS, as previously demonstrated in older women [43,44].

A previous study of 102 older adults from Brazil showed that higher physical performance is associated with lower body fat indicators and higher values of muscle mass and strength measurements [45]. However, the relationship of physical performance with anthropometric indicators of fat mass and muscle mass has been poorly studied. In this study, SPPB scores significantly decrease in the Q3 and Q4 quartiles of WCR. These findings may be related to the impact of visceral fat and other fat mass indexes on the reduced physical performance in older women reported in other studies [46,47,48].

Some limitations of these findings should be acknowledged. First, the sample consisted exclusively of older women, which can limit the generalizability of these findings to other populations. This limits the external validity of the findings, as they may not be generalizable to other populations, such as older men, individuals living in rural or high-income urban settings, or populations from different cultural or socioeconomic backgrounds. Thus, caution should be exercised when extrapolating these results beyond the study context. Second, the cross-sectional design precludes establishing causal relationships between the waist–calf circumference ratio and physical performance. Third, a reliable and validated measurement technique or other imaging method, like DXA or MRI, could provide a more precise assessment of body composition. Finally, unmeasured confounders, including dietary intake, hormonal levels, and inflammation, may have influenced these results.

Future research should consider confounders, such as dietary intake, hormonal levels, and inflammation, to enhance the robustness of the findings. Furthermore, a multisite study that captures older adults of different socioeconomic status and homogeneous samples of men and women is needed. Additionally, a longitudinal study is required to evaluate the temporal relationship between WCR and physical performance.

Our findings underscore the need for health approaches that integrate body composition, especially toward increased muscle mass and reduced visceral fat, as well as periodic assessments of muscle strength and physical function in older women. Timely identification of alterations in some of the aforementioned components through practical and easy-to-use indicators such as the WCR is essential in outpatient, community, and clinical settings due to limited accessibility to more complex tools. This study provides a useful starting point to advance the characterization and comorbidities associated with aging in the specific context of Latin America.

## 5. Conclusions

This study suggests the importance of the WCR, which was found to be significantly associated with indicators of body composition. Specifically, a high WCR and skeletal muscle mass (fat-free mass and skeletal muscle mass) are correlated; a high waist–calf circumference ratio and fat mass (appendicular skeletal muscle mass and visceral fat) are correlated; and a waist–calf circumference ratio and indicators of a high risk of sarcopenia (handgrip strength and SPPB) are correlated. Therefore, the WCR could be significantly useful for the rapid assessment of these body components in similar populations, especially in settings in which body composition measurement is limited. However, one limitation inherent to a cross-sectional study is that it cannot represent a temporal association with the study variables. Future studies are needed to confirm and extend the findings of this study to advance the characterization and comorbidities associated with aging in the specific context of Latin America.

## Figures and Tables

**Figure 1 geriatrics-10-00103-f001:**
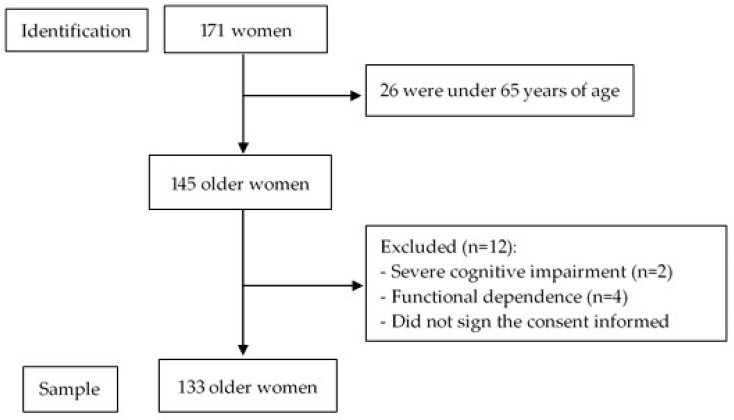
Flowchart of the recruitment process of the participants in the study.

**Figure 2 geriatrics-10-00103-f002:**
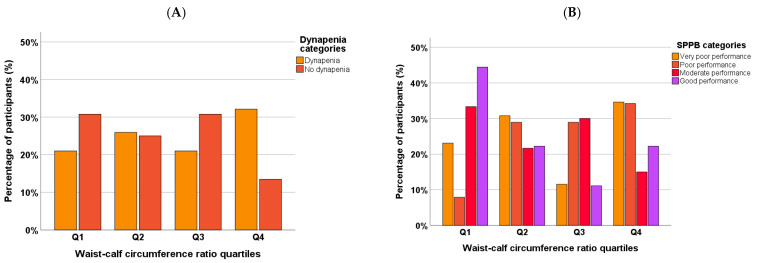
Categorical analysis of dynapenia (**A**) and Short Physical Performance Battery (SPPB) (**B**) categories according to waist–calf circumference ratio quartiles.

**Figure 3 geriatrics-10-00103-f003:**
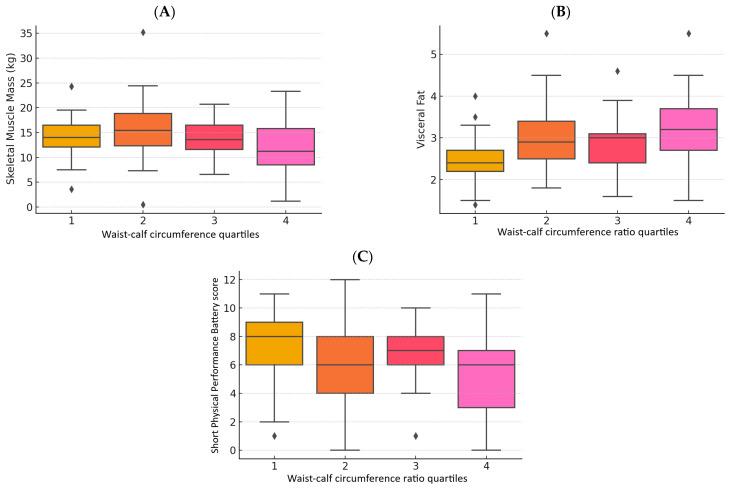
Boxplots of skeletal muscle mass (**A**), visceral fat (**B**), and Short Physical Performance Battery (SPPB) score (**C**) according to waist –calf circumference quartiles.

**Figure 4 geriatrics-10-00103-f004:**
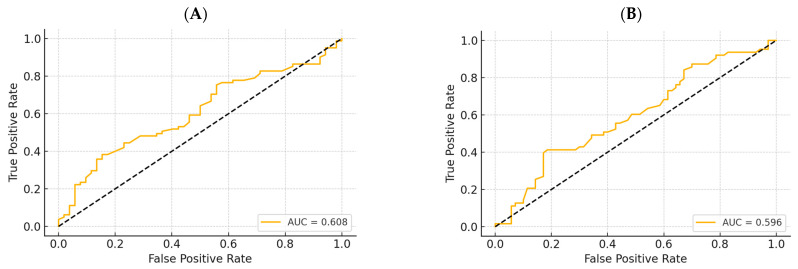
ROC curve for dynapenia (handgrip strength < 16 kg) (**A**) and low Short Physical Performance Battery score (<7) (**B**).

**Table 1 geriatrics-10-00103-t001:** Characteristics of the study participants.

Variable	*n*	%
Age, years	75 [IQR: 69.00–85.00]	
Ethnicity		
Mestizo	92	69.20
Afro-Ecuadorian	15	11.30
Caucasic	15	11.30
Indigenous	11	8.30
Marital status		
Single	40	30.10
Married	38	28.60
Widowed	44	33.10
Divorced	11	8.30
Instruction		
None	39	29.30
Elementary	73	54.90
High school	18	13.50
University	3	2.30
Height, cm	147.30 [IQR: 142.85–152.15]	
Weight, kg	60 [IQR: 51.80–66.50]	
BMI, kg/m^2^	27.42 ± 4.83	
Waist circumference, cm	92.37 [IQR: 90.50–94.24]	
Abdominal obesity		
Yes	90	67.70
No	43	32.30
Calf circumference, cm	33.20 [IQR: 29.85–35.90]	
Nutritional status		
Malnutrition	19	14.30
Nutritional risk	61	45.90
Normal	53	39.80
Frailty		
Yes	25	18.85
No	108	81.15
Sarcopenia risk		
Yes	63	47.40
No	70	52.60

BMI: body mass index.

**Table 2 geriatrics-10-00103-t002:** Indicators of body composition, muscle function, and strength according to waist–calf circumference ratio quartiles.

Variable	Total (*n* = 133)	Q1 (*n* = 33)	Q2 (*n* = 34)	Q3 (*n* = 33)	Q4 (*n* = 33)	*p* Value
FFM, kg	25.80 [9.45]	33.40 [6.85]	37.15 [11.92]	33.40 [9.90]	32.60 [10.45]	0.043 *
SMM, kg	14.10 [6.25]	14 [4.20]	15.45 [7.28]	13.60 [5.20]	11.20 [7.75]	0.039 *
ASM, kg	14.25 [3.62]	14.05 [2.70]	15.44 [4.89]	14.46 [3.48]	13.45 [3.72]	0.016 *
ASMI, kg/m^2^	6.52 [1.36]	6.50 [1.35]	7.09 [1.52]	6.38 [1.41]	6.28 [1.34]	0.051
Body fat, kg	25.80 [9.45]	25.40 [9.05]	25.90 [14.12]	26.60 [8.65]	24.90 [9.6]	0.535
BFI, kg/m^2^	11.60 [4.1]	11.40 [5.15]	11.60 [6.63]	12.20 [3.75]	10.90 [3.65]	0.568
Visceral fat, l	2.80 [1.00]	2.40 [0.60]	2.90 [0.95]	3.00 [0.85]	3.20 [1.10]	0.000 *
HGS, kg	13 [8.00]	14 [8.00]	14 [10.00]	14 [7.50]	10 [6.00]	0.025 *
SPPB value	7 [4.00]	8.00 [3.00]	7.00 [3.00]	6.00 [4.00]	6.00 [1.5]	0.029 *

* *p* < 0.05. All continuous variables are expressed as median [IQR] based on non-normal distribution. Q: quartile of WCR. FFM: fat-free mass; SMM: skeletal muscle mass; ASM: appendicular skeletal muscle mass; ASMI: appendicular skeletal muscle mass index; BFI: body fat index; HGS: handgrip strength; SPPB: Short Physical Performance Battery.

**Table 3 geriatrics-10-00103-t003:** Spearman correlations between waist–calf circumference ratio, body composition, and muscle function variables.

Variable	ρ (Spearman)	*p* Value
Skeletal muscle mass (kg)	−0.150	0.085
ASM (kg)	−0.198	0.022 *
ASMI (kg/m^2^)	−0.157	0.071
Handgrip strength (kg)	−0.235	0.006 *
SPPB	−0.156	0.074
Visceral fat	0.387	<0.001 *
Fat mass (kg)	−0.031	0.720
Body fat index (kg)	−0.147	0.092

* *p* < 0.05. ASM: appendicular skeletal muscle mass; ASMI: appendicular skeletal muscle mass index; SPPB: Short Physical Performance Battery.

## Data Availability

The datasets presented in this article are not readily available.

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
