# Peer review of "Waist–Calf Circumference Ratio Is Associated with Body Composition, Physical Performance, and Muscle Strength in Older Women"

_geriatrics, 2025, doi:10.3390/geriatrics10040103_

Round 1

Reviewer 1 Report

Comments and Suggestions for Authors

This manuscript addresses an important and emerging anthropometric index—waist-calf circumference ratio (WCR)—and its relationship with body composition and physical function in older women. The study is timely and of high clinical relevance, especially in low-resource settings. However, the manuscript requires improvements in structure, clarity, and methodological transparency before it can be considered for publication.

The choice of quartiles to categorize WCR is not clearly justified or supported by references. It is unclear whether these quartiles correspond to clinically meaningful thresholds. Provide a stronger justification for the use of quartiles (vs. cut-off points from ROC curves, previous studies, etc.).

The authors mention that no confounding variables were found in stratified analysis, but no details are given about which variables were tested or what the criteria were. Explicitly state which variables were tested as potential confounders and report the method used to assess confounding (e.g., stratification, multivariate model, interaction terms).

ANOVA and Kruskal-Wallis tests are used, but post-hoc comparisons (e.g., pairwise differences between WCR quartiles) are not presented despite evident gradients in outcomes. Add post-hoc pairwise tests with appropriate correction (e.g., Bonferroni or Dunn) and report effect sizes to clarify where significant differences lie.

Although mean/median values are reported, it would strengthen the findings to also categorize SPPB and HGS results based on validated cutoffs (e.g., dynapenia threshold, functional impairment). Include categorical analysis of HGS (<16kg) and SPPB (e.g., <7 points) by WCR quartile.

Table 2 is dense and difficult to interpret quickly. Use graphical representation (e.g., boxplots or bar graphs with error bars) for key results.  Provide sample sizes per group in the table headers. Consider a summary table highlighting key trends across WCR quartiles.

The discussion is repetitive and sometimes speculative without clearly linking results to prior findings. The conclusion overstates the implications of this cross-sectional study. Clearly delineate interpretation, implications, limitations, and recommendations for future research. Rephrase conclusions to avoid causal language, and reinforce the observational nature of the findings.

Comments on the Quality of English Language

Numerous grammatical errors, awkward phrasing, and redundancy hinder readability throughout the manuscript. The introduction and discussion contain long, unfocused paragraphs that need clearer topic sentences and logical flow. The manuscript needs thorough professional English editing to enhance clarity and academic tone.

Author Response

Response to Reviewer 1

Comment 1: This manuscript addresses an important and emerging anthropometric index—waist-calf circumference ratio (WCR)—and its relationship with body composition and physical function in older women. The study is timely and of high clinical relevance, especially in low-resource settings. However, the manuscript requires improvements in structure, clarity, and methodological transparency before it can be considered for publication.

Response 1: Thank you for pointing this out. We improved the structure, clarity, and methodological transparency.

Comment 2:  The choice of quartiles to categorize WCR is not clearly justified or supported by references. It is unclear whether these quartiles correspond to clinically meaningful thresholds. Provide a stronger justification for the use of quartiles (vs. cut-off points from ROC curves, previous studies, etc.).

Response 2: Since there are no validated clinical cutoff points for WCR in older women, it was decided to divide the sample into quartiles, which has been done in similar studies on anthropometric indicators. Citations of such studies have been added in Section 2.2.2.

Comment 3: The authors mention that no confounding variables were found in stratified analysis, but no details are given about which variables were tested or what the criteria were. Explicitly state which variables were tested as potential confounders and report the method used to assess confounding (e.g., stratification, multivariate model, interaction terms).

Response 3: The confounding variables were those that showed a significant association with the dependent variable and with WCR in a bivariate analysis. In this study, age, BMI, nutritional status, frailty, and risk of sarcopenia were determined as confounding variables. Linear and logistic regression models were constructed, adjusting for age, BMI, nutritional status, frailty, and risk of sarcopenia. This information was added to the Section 2.6.

Comment 4: ANOVA and Kruskal-Wallis tests are used, but post-hoc comparisons (e.g., pairwise differences between WCR quartiles) are not presented despite evident gradients in outcomes. Add post-hoc pairwise tests with appropriate correction (e.g., Bonferroni or Dunn) and report effect sizes to clarify where significant differences lie.

Response 4:  

We corrected the use of the test and only applied the Kruskal-Wallis test because all the analysis variables had a nonparametric distribution. We applied the Bonferroni correction for Kruskal–Wallis:

  • Participants in Q4 had lower HGS compared to those in Q1 (mean difference = 3.82 kg), with a moderate to large effect size (Cohen’s d = 0.672; r = 0.318). According to SMM, the comparison between Q1 and Q4 showed a moderate effect size (Cohen’s d = 0.483; r = 0.235), despite not reaching statistical significance (p = 0.066).
  • The Bonferroni post-hoc test showed significant differences between Q1 and Q3 (p = 0.018), Q1 and Q4 (p = < 0.001), and Q2 and Q4 (p = 0.006).
  • Regarding the SPPB scores according to WCR quartiles, it was found that Q1 vs. Q4 and Q2 vs. Q4 remained at p < 0.05, indicating that subjects in Q4 (highest WCR) had significantly lower SPPB scores than Q1 and Q2. Q1 vs. Q3 (p= 0.072) and Q2 vs. Q3 (p= 0.108) showed trends but did not reach the 0.05 threshold after correction.

These results are shown in Section 3.8.

Comment 5: Although mean/median values are reported, it would strengthen the findings to also categorize SPPB and HGS results based on validated cutoffs (e.g., dynapenia threshold, functional impairment). Include categorical analysis of HGS (<16kg) and SPPB (e.g., <7 points) by WCR quartile.

Response 5: Suggested analysis and chart added in Section 3.7.

Comment 6: Table 2 is dense and difficult to interpret quickly. Use graphical representation (e.g., boxplots or bar graphs with error bars) for key results.  Provide sample sizes per group in the table headers. Consider a summary table highlighting key trends across WCR quartiles.

Response 6:  To make Table 2 more readable, the interquartile range has been added to the summary and box plots have been created with the most relevant results. In addition, we used boxplots for skeletal muscle mass, Short Physical Performance Battery score and visceral fat. These results were updated in Section 3.6.

Comment 7: The discussion is repetitive and sometimes speculative without clearly linking results to prior findings. The conclusion overstates the implications of this cross-sectional study. Clearly delineate interpretation, implications, limitations, and recommendations for future research. Rephrase conclusions to avoid causal language, and reinforce the observational nature of the findings.

Response 7: Agreed, Sections 4 and 5 have been updated accordingly.

Comment 8: Numerous grammatical errors, awkward phrasing, and redundancy hinder readability throughout the manuscript. The introduction and discussion contain long, unfocused paragraphs that need clearer topic sentences and logical flow. The manuscript needs thorough professional English editing to enhance clarity and academic tone.

Response 8: We appreciate the reviewer’s valuable feedback regarding the manuscript’s language and structure. In response, the entire manuscript—particularly the Introduction and Discussion sections—has undergone extensive professional English editing to correct grammatical issues, improve sentence structure, and enhance clarity and academic tone. Long paragraphs have been revised to include clearer topic sentences and a more logical flow of ideas. Redundant content has been removed or rephrased to avoid repetition. We trust that these revisions have significantly improved the overall readability and quality of the manuscript.

Reviewer 2 Report

Comments and Suggestions for Authors

Title: ok

Abstract:
The index is not so new to be cited as such. The presentation of the result and conclusion can be more attractive and therefore enhance the value of the study, for this I suggest changing the objective

Introduction:
Check information on line 74 (WCR and waist circumference values, together with lower CC), if wcr is the perimeter of both, the sentence loses its meaning
Present studies that indicate how circumference can predict the variables analyzed (Body indicators included fat free mass (FFM), skeletal muscle mass (SMM), appendicular muscle mass (ASM), appendicular muscle mass index (ASMI), visceral fat (VF) and fat mass (FM), fat mass index (FMI). Handgrip strength (HGS) and the score of the Short Physical Performance Battery test (SPPB).

Methods
How the sample was selected: non-probabilistic
Since it excluded people with musculoskeletal diseases, over 65 it is inherent to AO and osteophytosis
Formatting
What is the purpose of asking about ethnicity and degree of Instructions:

Were the impedance device and the dynamometer calibrated before the research?

Perform correlation and regression tests to verify whether the WCR is a predictor of which variables.

Results
The sample had an average BMI that corresponds to overweight, how does this impact the findings and objectives of the study?

There is no caption for Q1-4 in the table.

The table does not indicate where the difference is by the anova test.

A statistical analysis is needed to compare the results between the variables.

Discussion
Add discussion regarding the new suggested analysis.

Conclusion
Redo after the analyses.

Author Response

Comment 1: Abstract:

The index is not so new to be cited as such. The presentation of the result and conclusion can be more attractive and therefore enhance the value of the study, for this I suggest changing the objective

Response 1: Thanks for the suggestion. We omit the term "novel" in the introduction of the summary.

Comment 2: Introduction:

Check information on line 74 (WCR and waist circumference values, together with lower CC), if wcr is the perimeter of both, the sentence loses its meaning

Present studies that indicate how circumference can predict the variables analyzed (Body indicators included fat free mass (FFM), skeletal muscle mass (SMM), appendicular muscle mass (ASM), appendicular muscle mass index (ASMI), visceral fat (VF) and fat mass (FM), fat mass index (FMI). Handgrip strength (HGS) and the score of the Short Physical Performance Battery test (SPPB).

Response 2: We have corrected this paragraph in section 1.

Comment 3: Methods

How the sample was selected: non-probabilistic

Since it excluded people with musculoskeletal diseases, over 65 it is inherent to AO and osteophytosis

Response 3: The sample was selected through non-probabilistic sampling applying inclusion and exclusion criteria. Section 2.1 was updated to include these details.

Comment 4: Formatting

What is the purpose of asking about ethnicity and degree of instruction:

Response 4: Ethnicity and educational level were collected as sociodemographic descriptors to better characterize the study population and explore potential differences in body composition, muscle function, or health risk profiles across subgroups. These variables are commonly associated with disparities in health outcomes, access to care, nutritional status, and lifestyle factors in older adults. While they were not the primary focus of the analysis, collecting this information allows for a more comprehensive understanding of the context and for potential future analyses on social determinants of health. In this study, we found no differences in the variables of interest between the subgroups.

Comment 5:  Were the impedance device and the dynamometer calibrated before the research?

Response 5: Yes, the equipment was calibrated before the research.

Comment 6:  Perform correlation and regression tests to verify whether the WCR is a predictor of which variables.

Response 6: Thank you for the recommendations. We have performed Spearman correlation tests and linear regressions. These results were included in Section 3.7.

Comment 7: Results The sample had an average BMI that corresponds to overweight, how does this impact the findings and objectives of the study?

Response 7: The fact that the sample had a mean BMI in the overweight range underscores the clinical relevance of using the waist-to-calf circumference ratio (WCR) to detect unfavorable body composition patterns. Overweight older adults may have adiposity excess while also presenting reduced muscle mass, a condition often undetected by BMI alone. Therefore, the observed associations between WCR, muscle strength, and function in this population support the utility of WCR as a more sensitive marker of age-related health risks compared to BMI. This discussion was added in Section 4.

Comment 8: There is no caption for Q1-4 in the table.

Response 8: We have added the description of the quartile abbreviations in Table 2.

Comment 9: The table does not indicate where the difference is by the anova test.

Response 9: Thank you for this observation. We agree with the reviewer that it is essential to specify where the significant differences occur. In response, we corrected the use of the test and only applied the Kruskal-Wallis test because all the analysis variables had a nonparametric distribution. The results of the post hoc comparisons Bonferroni-corrected Mann–Whitney U tests were included in Results section 3.7. These comparisons clarify the specific WCR quartiles that differ significantly for each variable.

Comment 10: A statistical analysis is needed to compare the results between the variables.

Response 10: In response, we have expanded the Results section to explicitly describe the statistical tests used to compare variables across WCR quartiles. For continuous variables, we applied Kruskal–Wallis test for non-normally distributed data. Where significant differences were found, post hoc analyses (Bonferroni-corrected Mann–Whitney U tests) were conducted to identify specific group differences. These results were included in Sections 3.6 and 3.8.

Comment 11: Discussion

Add discussion regarding the new suggested analysis.

Response 11: Agreed, Section 4 has been updated accordingly.

Comment 12: Conclusion

Redo after the analyses.

Response 12: Agreed, Section 5 has been updated accordingly.

Reviewer 3 Report

Comments and Suggestions for Authors

Firstly, thank you for this interesting research work. You make a clear case for why this metric should be explored and particularly in the population chosen for participation in this study. A good sample size is investigated.

A few specific considerations:

Line 60 - consider rephrasing/removing 'just to mention a few' it is overly informal.

Line 65 - I'm unsure of what you mean by limited mobility of bioimpedance devices, these are typically mobile and this is often reported as a benefit over more gold standard body composition analysis such as DEXA. Is this a geographical issue?

Line 85 - You state the objective is to explore relationships between WCR and body composition parameters but this isn't really what you have then gone on to do with your analysis and it also misses many of your variables that are more physical function related from this objective. Consider rewording.

Could you justify why you have excluded common comorbidities such as osteoarthritis from your sample?

Line 164- 170 - Why are two sets of interpretations given for SPPB scores? only one is then referred to within your results section.

Stats analysis and interpretation - As outlined above the main issue with your work in it's current form is the disconnect between your objective and the way in which you have analysed and then described your results. You discuss looking for relationships/associations but then have used quartiles to look for differences between groups rather than for relationships between variables. This needs addressing. 

Author Response

Firstly, thank you for this interesting research work. You make a clear case for why this metric should be explored and particularly in the population chosen for participation in this study. A good sample size is investigated.

Comment 1: Line 60 - consider rephrasing/removing 'just to mention a few' it is overly informal.

Response: 1: Thank you for the comment; we have removed the sentence.

Comment 2: Line 65 - I'm unsure of what you mean by limited mobility of bioimpedance devices, these are typically mobile and this is often reported as a benefit over more gold standard body composition analysis such as DEXA. Is this a geographical issue?

Response 2: We agree that bioimpedance devices are generally considered more portable and accessible than gold-standard methods such as DEXA. Our original wording may have been misleading. What we intended to convey is that: in certain low-resource or community-based settings, such as in some regions of Latin America, access to even portable BIA devices can be limited due to their cost, maintenance needs, or lack of trained personnel. We have revised the text to clarify that the limitation is related to availability and accessibility in specific primary care or geographic contexts, rather than the physical portability of the devices themselves.

Comment 3: Line 85 - You state the objective is to explore relationships between WCR and body composition parameters but this isn't really what you have then gone on to do with your analysis and it also misses many of your variables that are more physical function related from this objective. Consider rewording.

Response 3:  We agree that the original wording of the study objective did not fully reflect the scope of the variables analyzed, particularly those related to muscle strength and physical function. In response, we have revised the objective in the Abstract and Introduction to better capture the comprehensive nature of our analysis, which included indicators of body composition, muscle strength, and physical performance. We believe this revised wording more accurately aligns with the study's methodology and findings.

Comment 4: Could you justify why you have excluded common comorbidities such as osteoarthritis from your sample?

Response 4: Osteoarthritis and other common comorbidities were excluded to minimize potential confounding effects on muscle function and mobility assessments, particularly in variables such as gait speed, chair stands, and handgrip strength. Because the primary objective was to assess associations between WCR and physical function and strength, we aimed to reduce variability introduced by musculoskeletal conditions that may independently impair performance regardless of body composition. This information was included in Section 2.1.

Comment 5: Line 164- 170 - Why are two sets of interpretations given for SPPB scores? only one is then referred to within your results section.

Response 5: We agree that including two sets of SPPB score interpretations without clarifying their use may cause confusion. Initially, we presented both classification systems to provide context, as both are reported in the literature. However, only the four-category interpretation was used for the analysis and reported in the results. To avoid confusion, we have removed the unused classification and retained only the scoring system that aligns with our analysis throughout the manuscript.

Comment 6: Stats analysis and interpretation - As outlined above the main issue with your work in it's current form is the disconnect between your objective and the way in which you have analysed and then described your results. You discuss looking for relationships/associations but then have used quartiles to look for differences between groups rather than for relationships between variables. This needs addressing.

Response 6: We acknowledge the discrepancy between the stated objective—to explore associations between WCR and key health indicators—and the primary analytical approach, which focused on comparing quartiles of WCR. In response, we have revised the objective throughout the manuscript to better reflect both the correlation and group comparison analyses. Additionally, we have clarified in the Methods and Results sections that our analysis included both (1) correlation and regression models to assess continuous associations, and (2) comparisons across WCR quartiles to identify group-level differences. This dual approach was intended to provide a comprehensive view of the relationships between WCR and body composition, strength, and function, and is now clearly explained in the revised manuscript.

Round 2

Reviewer 1 Report

Comments and Suggestions for Authors

The revised version clearly and comprehensively addresses all the concerns raised by the reviewer. The authors demonstrated diligence in methodological refinement, statistical rigor, visual data presentation, and professional editing. These improvements substantially elevate the manuscript’s scientific value and readability.

Although the revised manuscript has significantly improved, a few minor issues remain that could enhance its clarity and scientific rigor. First, the Limitations section, while more robust, would benefit from an explicit discussion of external validity concerns. Specifically, the findings are based on a sample of urban Ecuadorian older women, and caution should be exercised when generalizing these results to other populations, such as men, rural communities, or different cultural and socioeconomic contexts.

Second, the visual figures, although well-conceived, should be carefully reviewed for consistency in labeling and axis scaling. Uniformity in graphical presentation is crucial for maintaining clarity, especially for readers interpreting comparative trends across variables.

Lastly, the manuscript exhibits occasional inconsistency in the use of the terms “mean” and “median” when summarizing data. Standardizing the terminology throughout the text and tables, based on the distribution of variables, would further improve the precision and coherence of the presentation. Addressing these minor issues will help ensure the manuscript meets the highest standards of scientific communication.

Author Response

Comment 1: The revised version clearly and comprehensively addresses all the concerns raised by the reviewer. The authors demonstrated diligence in methodological refinement, statistical rigor, visual data presentation, and professional editing. These improvements substantially elevate the manuscript’s scientific value and readability.

Response 1: We appreciate the reviewer’s valuable feedback regarding the manuscript’s language and structure.

Comment 2: Although the revised manuscript has significantly improved, a few minor issues remain that could enhance its clarity and scientific rigor. First, the Limitations section, while more robust, would benefit from an explicit discussion of external validity concerns. Specifically, the findings are based on a sample of urban Ecuadorian older women, and caution should be exercised when generalizing these results to other populations, such as men, rural communities, or different cultural and socioeconomic contexts.

Response 2: Thanks for the suggestion. In the limitation section we added “This limits the external validity of the findings, as they may not be generalizable to other populations, such as older men, individuals living in rural or high-income urban settings, or populations from different cultural or socioeconomic backgrounds. Thus, caution should be exercised when extrapolating these results beyond the study context.”

Comment 3: Second, the visual figures, although well-conceived, should be carefully reviewed for consistency in labeling and axis scaling. Uniformity in graphical presentation is crucial for maintaining clarity, especially for readers interpreting comparative trends across variables.

Response 3: Figures were revised and corrected as it shown in the results section.

Comment 4: Lastly, the manuscript exhibits occasional inconsistency in the use of the terms “mean” and “median” when summarizing data. Standardizing the terminology throughout the text and tables, based on the distribution of variables, would further improve the precision and coherence of the presentation. Addressing these minor issues will help ensure the manuscript meets the highest standards of scientific communication.

Response 4: We found an error in the discussion section related with terms “mean and median” and it was addressed.
